# Metformin Alleviates Inflammation and Induces Mitophagy in Human Retinal Pigment Epithelium Cells Suffering from Mitochondrial Damage

**DOI:** 10.3390/cells13171433

**Published:** 2024-08-27

**Authors:** Maija Toppila, Sofia Ranta-aho, Kai Kaarniranta, Maria Hytti, Anu Kauppinen

**Affiliations:** 1School of Pharmacy, Faculty of Health Sciences, University of Eastern Finland, 70211 Kuopio, Finland; sofia.ranta-aho@uef.fi (S.R.-a.); maria.hytti@uef.fi (M.H.); 2Department of Ophthalmology, Kuopio University Hospital, 70211 Kuopio, Finland; kai.kaarniranta@uef.fi; 3Department of Ophthalmology, Institute of Clinical Medicine, University of Eastern Finland, 70211 Kuopio, Finland

**Keywords:** metformin, AMD, inflammation, autophagy, RPE, mitochondrial ROS

## Abstract

Mitochondrial malfunction, excessive production of reactive oxygen species (ROS), deficient autophagy/mitophagy, and chronic inflammation are hallmarks of age-related macular degeneration (AMD). Metformin has been shown to activate mitophagy, alleviate inflammation, and lower the odds of developing AMD. Here, we explored the ability of metformin to activate mitophagy and alleviate inflammation in retinal pigment epithelium (RPE) cells. Human ARPE-19 cells were pre-treated with metformin for 1 h prior to exposure to antimycin A (10 µM), which induced mitochondrial damage. Cell viability, ROS production, and inflammatory cytokine production were measured, while autophagy/mitophagy proteins were studied using Western blotting and immunocytochemistry. Metformin pre-treatment reduced the levels of proinflammatory cytokines IL-6 and IL-8 to 42% and 65% compared to ARPE-19 cells exposed to antimycin A alone. Metformin reduced the accumulation of the autophagy substrate SQSTM1/p62 (43.9%) and the levels of LC3 I and II (51.6% and 48.6%, respectively) after antimycin A exposure. Metformin also increased the colocalization of LC3 with TOM20 1.5-fold, suggesting active mitophagy. Antimycin A exposure increased the production of mitochondrial ROS (226%), which was reduced by the metformin pre-treatment (84.5%). Collectively, metformin showed anti-inflammatory and antioxidative potential with mitophagy induction in human RPE cells suffering from mitochondrial damage.

## 1. Introduction

Metformin (dimethylbiguanide) is one of the most widely used drugs for the treatment of type II diabetes mellitus. Metformin was introduced to the public in 1957, but already in medieval Europe, *Galega officinalis*, the herbal predecessor of metformin, was used to treat various illnesses [1]. To cite the pharmacologist Sir James Whyte Black, “The most fruitful basis for the discovery of a new drug is to start with an old drug” [2]. Metformin has proven to be a good candidate for alternative uses; in addition to diabetes, it is used to treat polycystic ovary syndrome [PCOS; [3,4]] and obesity [4], and its properties in treating cancer are under investigation [4,5,6]. Interestingly, population-based studies have revealed that patients receiving metformin are less likely to develop age-related macular degeneration [AMD; [7,8,9,10,11]].

As one of the most common eye diseases amongst the elderly, it is estimated that over 200 million people will be suffering from AMD by 2030 and up to 300 million by the year 2040 [12]. AMD damages the retinal pigment epithelium cells (RPE) in the macula at the center of the retina, the area that is responsible for the sharp and central vision [13]. Several investigators have shown that prolonged inflammation, oxidative stress, mitochondrial damage, and impaired mitophagy are associated with the pathology of AMD [14,15]. Interestingly, mitochondrial damage itself triggers an increase in oxidative stress and cellular inflammation [15]. Mitophagy is a selective form of macroautophagy (hereafter autophagy) in which damaged mitochondria are encapsulated and degraded by lysosomal enzymes. Previous studies have shown the potential of metformin to activate mitophagy [16,17].

In this study, we explored the effects of metformin in human RPE cells suffering from mitochondrial damage. We hypothesized that metformin-induced activation of mitophagy and the subsequent removal of damaged mitochondria might reduce RPE cell stress by limiting oxidative stress and proinflammatory signaling.

## 2. Materials and Methods

### 2.1. Cell Culture

The experiments were conducted using the ARPE-19 cell line, which was purchased from the American Type Culture Collection (ATCC, passage 19, Manassas, VA, USA). The cells were cultured in a 1:1 mixture of Dulbecco’s modified Eagle’s medium (DMEM) and nutrient mixture F-12 (Life Technologies, Carlsbad, CA, USA) supplemented with 10% HyClone™ fetal bovine serum (FBS; Thermo Fisher Scientific, Waltham, MA, USA), 100 U/mL penicillin, 100 μg/mL streptomycin (both Lonza, Basel, Switzerland or Life Technologies, Carlsbad, CA, USA), and 2 mM L-glutamine (Life Technologies, Carlsbad, CA, USA). The cells were passaged routinely every 3–4 days using 0.25% Trypsin–EDTA (Life Technologies, Carlsbad, CA, USA) and used in experiments at cumulative passages between 26–35. The cells were maintained in an incubator providing a humidified atmosphere at +37 °C with 5% CO_2_.

### 2.2. Cell Treatments

In the experiments, the cells were plated on 12-well plates (viability assays, enzyme-linked immunosorbent assays, Western blot analyses) at a density of 200,000 cells/well, on 96-well plates (ROS, mitoSOX) at a density of 15,000 cells/well, on 6-well plates (RNA extraction for qPCR) at a density of 500,000 cells/well, or on 8-well chamber slides (Lab-Tek™ Chamber Slide System, Thermo Fisher Scientific, Rochester, NY, USA; immunofluorescence, mitoTracker staining) at a density of 30,000 cells/well and cultured for three days to reach confluency. All experiments were performed in serum-free culture medium. Cells were exposed to different concentrations of metformin (10–40 mM) in the initial cell viability assays and assessments of their anti-inflammatory effects before selecting the concentration of 15 mM metformin to be used in the subsequent experiments. Metformin treatment occurred 1 h prior to the exposure to 10 µM antimycin A for an additional 2–24 h. Control cells were exposed to DMSO at equimolar percentages in comparison to the respective treatment group.

### 2.3. Cell Viability Assays

ARPE-19 cells were exposed to metformin for 24 h or for 1 h, followed by exposure to antimycin A (10 µM) for 24 h. Subsequently, cell viability was determined by measuring lactate dehydrogenase (LDH) from fresh medium samples according to the manufacturer’s protocol (CytoTox96^®^ Non-Radioactive Cytotoxicity Assay, Promega, Madison, WI, USA). Cellular viability was also detected using the 3-(4,5-dimethylthiazol-2-yl)-2,5-diphenyltetrazolium bromide (MTT, Sigma-Aldrich, St. Louis, MO, USA) assay. MTT salt solution was added to the cell culture medium at the final concentration of 500 µg/mL, and the cells were incubated for 90 min at +37 °C with protection from light. After the incubation, the medium was removed, and formazan crystals were dissolved in DMSO (Fischer Scientific, Leics, UK). Absorbance values were measured using a spectrophotometer (BioTek, ELx808, Microplate reader with the Gen-5 2.04 program, BioTek Instruments Inc., Winooski, VT, USA) at a wavelength of 562 nm.

### 2.4. Enzyme-Linked Immunosorbent Assay (ELISA)

The levels of proinflammatory cytokines IL-6 and IL-8 were measured from medium samples using BD OptEIA^TM^ Human ELISA Kits (BD Biosciences, San Diego, CA, USA) according to the manufacturer’s protocols. Medium samples were collected after cells were exposed to metformin (10, 15, 20 mM) for 1 h and antimycin A (10 µM) for 24 h. Absorbance values were measured using a spectrophotometer (BioTek, ELx808, Microplate reader with the Gen-5 2.04 program, BioTek Instruments Inc., Winooski, VT, USA) at a wavelength of 450 nm with a reference wavelength of 620 nm. The levels of phosphorylated mitogen-activated protein kinases (MAPKs) p38, ERK1/2, and JNK1 were measured from cell lysates following exposure to metformin (15 mM) for 1 h and antimycin A (10 µM) for 2 h, using human PathScan^®^ ELISA Kits (Cell Signalling Technologies, Beverly, MA, USA), while the DNA binding activity of the NF-κB subunit p65 was measured at the same timepoint using the TransAM^®^ NFkB p65 ELISA Kit (Active Motif, Carlsbad, CA, USA). All measurements were performed according to the manufacturers’ instructions.

### 2.5. Western Blot

The Western blot method was used to detect the autophagy markers p62/SQSTM1 and LC3. Our autophagy assays and interpretation are based on “Guidelines for the use and interpretation of assays for monitoring autophagy” by Klionsky and colleagues [18]. ARPE-19 cells were treated with metformin (15 mM) for 1 h prior to exposure to antimycin A (10 µM) for 6 h. Subsequently, the culture medium was removed, and the cells were lysed with the mammalian protein extraction reagent (M-PER, Thermo Fisher Scientific, Waltham, MA, USA) according to the manufacturer’s instructions. Protein concentrations were determined from the lysates using the Bradford protocol [19]. Samples containing 30–40 µg of protein were loaded onto 15% SDS-PAGE gels. Protein bands were separated at 200 V and wet-blotted onto nitrocellulose membranes (GE Healthcare, Little Chalfont, Buckinghamshire, UK) overnight at 17 V with a blotting buffer containing 20% methanol. Protein transfer was confirmed using the Ponceau S (Sigma-Aldrich, St. Louis, MO, USA) solution. The membranes were blocked with 3% milk in 0.3% Tween-PBS (Sigma-Aldrich, St. Louis, MO, USA) for 1.5 h at RT. After blocking, the membranes were washed for 3 × 5 min with the primary antibody dilution buffer without milk or bovine serum albumin (BSA). The washing steps were also repeated between primary and secondary antibodies before the addition of the substrate solution (Millipore, Burlington, MA, USA). Primary antibodies for p62/SQSTM1 (1:1000 in 0.5% BSA in 0.3% Tween-PBS; sc-28359; Santa Cruz Biotechnology, Santa Cruz, CA, USA) or LC3 (1:1000 in 0.1% Tween-TBS; AP1802a; Abgent, San Diego, CA, USA) were incubated overnight at +4 °C. The primary antibody for GAPDH (1:15,000 in 0.1% Tween-PBS; ab8245; Abcam, Cambridge, UK) was incubated for 2 h at RT. Anti-mouse secondary antibody (NA931; GE Healthcare, Chicago, IL, USA) for p62/SQSTM1 (1:10,000 in 3% milk in 0.3% Tween-PBS) was incubated for 2 h at RT and for GAPDH (1:12,000 in 0.1% Tween-PBS) for 1 h at RT. Anti-rabbit secondary antibody for LC3 (1:5000 in 3% milk/0.1% Tween-TBS; A16104; Invitrogen, Carlsbad, CA, USA) was incubated for 2 h at RT. The membranes were exposed to the Immobilon^®^ Western Chemiluminescent HRP Substrate (Millipore, Billerica, MA, USA), and protein bands were detected using the Image Quant RT ECL-camera (GE Healthcare, Little Chalfont, UK). Relative densities of bands from images were quantified using the ImageJ software (1.47v, US National Institutes of Health, Bethesda, MD, USA; http://imagej.nih.gov/ij/ (accessed 24 August 2024) and normalized to GAPDH.

### 2.6. Real-Time Polymerase Chain Reaction (PCR)

The Nucleospin^®^ RNA Plus kit (Macherey-Nagel, Düren, Germany) was used according to the manufacturer’s protocol to extract RNA from cells exposed to metformin (15 mM) for 1 h and to antimycin A (10 µM) for 3 h. RNA concentrations were measured using a NanoDrop^®^ spectrophotometer (Thermo Fisher Scientific, Waltham, MA, USA), and cDNA was synthesized using the SuperScript™ VILO™ cDNA synthesis kit (Invitrogen, Waltham, MA, USA). The levels of SQSTM1/p62, LC3, mTOR, and Nrf2 (NFE2L2) mRNA were analyzed using the Applied Biosystems QuantStudio^TM^ 5 Real-Time PCR system (Applied Biosystems by Life Technologies Europe BV) using the SYBR^®^ Green chemistry (Applied Biosystems by Life Technologies, Europe BV), as previously described [20]. The primers used (all from Metabion, Planegg/Steinkirchen, Germany) are shown in Table 1.

The relative mRNA levels of p62/SQSTM1, LC3, mTOR, and Nrf2 (NFE2L2) were normalized to those of GAPDH, which served as an endogenous control. Data were analyzed using the ΔΔC_T_ (change in cycle threshold, C_T_) method [21].

### 2.7. Immunofluorescence

Immunofluorescence was used to study mitophagy in ARPE-19 cells. Cells were treated with 15 mM metformin for 1 h prior to exposure to 10 µM antimycin A for 24 h. Thereafter, the cells were fixed for 15 min in 4% paraformaldehyde in PBS at RT and for another 15 min in ice-cold methanol at RT and washed 3 times between each step with Dulbecco’s Phosphate-Buffered Saline (DPBS, Life Technologies, Paisley, UK). Next, the cells were permeated with 3% BSA and 0.3% Triton-X in PBS for 1 h at RT. The primary antibodies for LC3 (see Western blot chapter) and TOM20 (sc-136211, Santa Cruz Biotechnology, Heidelberg, Germany) were diluted 1:200 in 1% BSA/0.1% Triton-X and incubated overnight at 4 °C. The secondary antibodies for LC3 (red, Alexa Fluor 594, goat anti-rabbit, A11037, Invitrogen, Carlsbad, CA, USA) and TOM20 (green, Alexa Fluor 488, goat anti-mouse, A11029, Invitrogen, Carlsbad, CA, USA) were also diluted 1:200 in 1% BSA/0.1% Triton-X and incubated for 1 h at RT. In order to stain the nuclei, the cultured cells were incubated with a 1:10,000 dilution of DAPI (D9541, Sigma, St. Louis, MO, USA) before mounting the cells in Mowiol mounting medium (Sigma, St. Louis, MO, USA). Three images per sample were taken at 63X magnification using a fluorescence microscope (Zeiss LSM700 confocal microscope, Göttingen, Germany). Fiji (ImageJ 1.54f) [22] was used to determine the threshold Manders colocalization coefficient [23] to estimate the colocalization of LC3 and TOM20 in three pictures/frames per repetition in each experiment.

### 2.8. MitoTracker Red Staining

The mitochondrial membrane potential (MMP) sensitive dye MitoTracker red CMX-ROS (Thermo Fisher) was utilized to assess the quality of the mitochondrial network. The ARPE-19 cells were treated with 15 mM metformin 1 h prior to 10 µM antimycin A exposure for 24 h. Thereafter, the cells were exposed to 200 nM mitoTracker Red for 15 min before Hoechst staining (NucBlue™ Live Cell Stain ReadyProbes™ reagent, Eugene, OR, USA) of nuclei for 10 min. Next, the cells were fixed with paraformaldehyde (PFA), washed, and mounted with the Mowiol mounting medium, as described above. Three images per sample were taken at a 65× magnification in a fluorescence microscope (Zeiss AX10 Imager A2, Zeiss, Göttingen, Germany) and analyzed using the ImageJ software.

### 2.9. Mitochondrial and Total Cellular ROS Levels

MitoSOX^TM^ Red (Invitrogen) was used to determine mitochondrial ROS levels. ARPE-19 cells were treated either with 15 mM metformin or 50 µM mitoTEMPO (ENZO Life Sciences, Farmingdale, NY, USA) and incubated for 1 h before exposure to 10 µM Aa for an additional 2 h. After the incubation, 2.5 µM mitoSOX was added, and cells were incubated for 10 min. Thereafter, the cells were washed three times with DPBS, and fresh DPBS was added. The fluorescence intensity (ex/em = 510/580 nm) was measured using the BioTek Cytation3 reader (BioTek, Instruments Inc., Winooski, VT, USA).

In order to measure total cellular ROS levels, ARPE-19 cells were exposed to 5 µM of the ROS-sensitive fluorescent dye 2′,7′-dichlorodihydrofluorescein diacetate (H2DCFDA, Life Technologies, Eugene, OR, USA) and 15 mM metformin for 1 h prior to the addition of 10 µM antimycin A for another hour. After incubation, the cells were washed twice with DPBS before fresh DPBS was added. Fluorescence intensity (ex/em = 485/530 nm) was measured using the BioTek Cytation3 imaging reader.

### 2.10. Statistical Analysis

Statistical analyses were performed using GraphPad Prism, version 9 (GraphPad Software, San Diego, CA, USA). The Mann–Whitney U-test was used for pairwise comparisons. *p*-values of <0.05 were considered statistically significant. The data are shown in the figures as means with standard error of the mean (SEM). All experiments were performed at least three times, days, weeks, or months apart from each other.

## 3. Results

### 3.1. ARPE-19 Cells Remain Viable after an Exposure up to 40 mM Metformin

According to the ISO 10993-5 standard [24], cells are considered viable as long as cellular viability remains above 70%. The tolerability of ARPE-19 cells toward metformin was assessed from the MTT and LDH assays. Cell viability remained over 80% at all measured concentrations (10 mM, 90%; 15 mM, 98%; 20 mM, 97%; 30 mM, 95%; 35 mM, 87%; 40 mM, 87%; Figure 1A). LDH leakage from the cells was significantly increased at or above the 25 mM metformin concentration (Figure 1B). Based on these results, concentrations of 10–20 mM were selected for further experiments.

### 3.2. Metformin Reduces the Levels of Proinflammatory Cytokines IL-6 and IL-8 in ARPE-19 Cells with Induced Mitochondrial Damage

The mitochondrial damage induced by antimycin A increased the production of both IL-6 (Figure 2A) and IL-8 (Figure 2B), though the increase in IL-8 was not statistically significant. All metformin concentrations (10 mM, 15 mM, and 20 mM) significantly reduced the cytokine levels, supporting the anti-inflammatory potential of metformin. Antimycin A was dissolved and diluted in DMSO, and an identical amount of DMSO was utilized as a vehicle control. The vehicle control increased LDH levels in the medium (Figure 2C). In our preliminary experiments, Antimycin A treatment did not cause LDH leakage from the ARPE-19 cells at a concentration of 10 µM (Figure 2C), and therefore, this concentration was utilized in future experiments. Metformin pre-treatment did not affect the levels of LDH in cells exposed to Aa (Figure 2C). Based on these results, we selected 15 mM metformin for further experiments.

### 3.3. Anti-Inflammatory Effects of Metformin Are Not Mediated by the Regulation of NF-κB or MAP Kinases

In order to study the mechanisms underlying the anti-inflammatory effects of metformin, we measured the DNA binding activity of nuclear factor kappa B (NF-κB) subunit p65 and the levels of phosphorylated MAP kinases (MAPKs) JNK, p38, and ERK1/2. These proteins are often associated with inflammation in RPE cells [25,26,27]. Antimycin A alone reduced the DNA binding activity of the NF-κB subunit p65, but this was not affected by the metformin pre-treatment (Figure 3A). Additionally, metformin had no significant effect on the phosphorylation state of any of the measured MAPKs (Figure 3B–D), suggesting that the anti-inflammatory effect of metformin was not being mediated via the NF-κB or MAPK pathways.

### 3.4. Metformin Pre-Treatment Prevents Antimycin A-Induced Loss of the Mitochondrial Membrane Potential in ARPE-19 Cells

To assess the quality of the mitochondrial network and the mitochondrial membrane potential (MMP), ARPE-19 cells were stained with mitoTracker Red CMX-ROS and analyzed under a confocal microscope. Antimycin A caused a significant loss of MMP, which was visible as a reduced fluorescence intensity when viewed in the microscope (Figure 4A). Pre-treatment with metformin significantly protected cells from the antimycin A-related reduction of MMP. Interestingly, metformin alone also significantly reduced the MMP (Figure 4). However, it prevented a further loss of MMP upon antimycin A exposure (Figure 4).

### 3.5. Metformin Affects the Levels of Autophagy Marker Proteins in ARPE-19 Cells

The levels of the autophagy substrate p62/SQSTM1 and LC3 were measured using the Western blot method. Antimycin A increased the levels of p62/SQSTM1 (Figure 5A) and induced the lipidation of LC3 I to LC3 II, as expected on the basis of our prior observations [Figure 5B,C [28]]. The exposure of RPE cells to metformin (15 mM) reduced the levels of p62/SQSTM1 (Figure 5A), LC3I (Figure 5B), and LC3II (Figure 5C), indicative of potentially enhanced clearance of accumulated autophagosomes.

Antimycin A treatment increased the transcription of SQSTM1 (Figure 6A) and MAP1LC3B (Figure 6B), but metformin had no effect on the measured mRNA levels, suggesting that the changes observed in Western blotting were related to protein clearance instead of protein synthesis. Neither metformin nor antimycin A affected the transcription of MTOR (Figure 6C).

### 3.6. Metformin Induced Mitophagy in ARPE-19 Cells

We used immunofluorescence to study ongoing mitophagy in the cells. TOM20 was used to label mitochondria and LC3 to identify autophagosomes. DAPI was used to stain the nuclei—this technique makes it feasible to determine the cell number per image. The pre-treatment of human RPE cells with metformin prior to exposure to antimycin A significantly increased the colocalization of LC3 and TOM20 (Figure 7), pointing to the potential of metformin to activate mitophagy in ARPE-19 cells.

### 3.7. Metformin Selectively Reduces Antimycin A-Induced Mitochondrial ROS Production in ARPE-19 Cells

Antimycin A increased the production of total cellular ROS, which was determined using the DCFDA dye (Figure 8A). Metformin pre-treatment had no additional effect (Figure 8A). Antimycin A treatment also increased the ROS production by mitochondria (Figure 8B), which was measured using the mitoSOX^TM^ Red reagent, and the pre-treatment of RPE cells with metformin reduced the levels of mitochondrial ROS by 16%. The effect of metformin was comparable to that of MitoTEMPO, a known mitochondria-targeted antioxidant that reduced the levels of mitochondrial ROS by 20% (Figure 8B).

Cellular ROS levels are controlled by Nrf2, “the master regulator of the antioxidant response”, which has an important role in mediating antioxidant functions during inflammation [29]. To study whether increased ROS led to Nrf2 activation, we used RT-qPCR to assess the levels of the Nrf2 gene, NFE2L2. In this study, both metformin and antimycin A increased the transcription of NFE2L2 (Figure 8C), suggesting that Nrf2 might be activated by antimycin A-induced ROS production and inflammation.

## 4. Discussion

Several studies have revealed the potential of metformin to improve mitochondrial respiration and autophagy, as well as alleviate mitochondrial ROS production and inflammation [30,31]. In the present study, metformin pre-treatment significantly reduced the secretion of two proinflammatory cytokines, IL-6 and IL-8, from human RPE cells suffering from mitochondrial damage. The reduced production of proinflammatory cytokines, such as IL-1β, IL-6, IL-8, TNF-α, (IL)-17A, or interferon (IFN)-γ, after metformin treatment has been previously demonstrated in different cell types, e.g., in peripheral blood mononuclear cells, T lymphocytes, mouse macrophages, endometriotic stromal cells, and ARPE-19 cells [16,32,33,34,35,36]. It has also been shown that patients undergoing metformin treatment experience a lower incidence of AMD [7,8,9,10,11]. Additionally, a phase II clinical trial to determine whether metformin can decelerate the progression of dry AMD in non-diabetic patients is ongoing (NCT02684578).

Several pathways have been found to mediate the anti-inflammatory effects of metformin. As an inhibitor of mitochondrial electron transport chain (ETC) complex 1, metformin exposure increases the intracellular AMP:ATP ratio. This imbalance activates AMPK-dependent pathways, such as NF-κB, mTOR, Nrf2, or NLRP3, or AMPK-independent signaling, such as STAT3, mTOR, hypoxia-inducible factor-1α (HIF-1α), or SIRT1 pathways [31]. Here, we measured the activity of MAPKs and NF-κB and the RNA levels of NFE2L2 to reveal the pathway(s) underlying the anti-inflammatory actions of metformin in human RPE cells. Antimycin A increased the DNA-binding activity of NF-κB and NFE2L2 mRNA levels, suggesting that NF-κB and/or Nrf2 might be involved in antimycin A-induced inflammation. However, pre-treatment with metformin did not alter the levels of NF-κB, NFE2L2 mRNA, or phosphorylated MAPKs in antimycin A-treated cells, indicating that these proteins are not responsible for the effects of metformin in our model. By itself, metformin increased the transcription of Nrf2, and further studies, e.g., exploring the translocation of Nrf2 to the nucleus, will be required to fully clarify the role of Nrf2 in the anti-inflammatory actions of metformin. In line with our results, studies conducted by Feng et al. demonstrated that by activating Nrf2 signaling, metformin reduced the production of IL-1β, IL-6, IL-8, and TNF-α in ARPE-19 cells treated with hydrogen peroxide (H_2_O_2_) [33].

There are several reports that metformin can either have antioxidative properties [37,38] or induce oxidative stress by causing mitochondrial malfunction [39]. In the present study, antimycin A treatment significantly increased both total cellular as well as mitochondrial ROS production in human RPE cells while metformin pre-treatment reduced mitochondrial ROS levels. Even though ROS are important signaling molecules, their excessive production can cause serious damage within the cell. Zhao et al. have reported that metformin reduced H_2_O_2_-induced ROS production and restored MMP in retinal D407 cells [40]. This is in line with our results showing that metformin protected ARPE-19 cells from antimycin A-induced MMP loss, even though it alone reduced MMP, likely via inhibition of complex 1 of the ETC.

The loss or lowering of MMP is a well-known trigger of mitophagy, and metformin has been found to activate mitophagy in retinal ganglion cells [17] and in peripheral blood mononuclear cells [16]. We found that antimycin A exposure increased the accumulation of p62/SQSTM1 and LC3-II in RPE cells, suggesting that while early-stage autophagy is activated after an antimycin A exposure [28], autophagosomes are inefficiently cleared. When the cells were exposed to metformin prior to the antimycin A treatment, this resulted in lower levels of both p62/SQSTM1 and LC3-II, an indication of either the activation of autophagic clearance or a reduced formation of autophagosomes. The increased autophagy is supported by the observation that metformin increased TOM20/LC3 colocalization, which suggests increased mitophagy in our cells. This is in line with the previous publication from Shu et al., who detected an increased autophagic flux upon metformin treatment in ARPE-19 cells treated with vital dyes, such as indocyanine green or brilliant blue G [41]. Similarly, Zhao et al. have demonstrated autophagy activation by metformin in the D407 RPE cell line via the activation of AMPK [40]. In this study, we did not focus on determining the pathway underlying autophagy/mitophagy activation since metformin has been described to work via several pathways. These include AMPK-dependent pathways, such as AMPK/mTOR or AMPK/ULK1, and AMPK-independent pathways, such as Redd1/mTOR or MAPK/ERK [42,43]. We did not observe changes in mTOR transcription levels, but this does not entirely rule out the involvement of mTOR, and the precise pathways regulating autophagy/mitophagy in RPE cells suffering from mitochondrial damage will need to be confirmed in future studies. Further studies will also be required to monitor the autophagic flux in metformin-treated RPE cells more comprehensively.

Overall, our data support the view that metformin reduced mitochondrial ROS production potentially via increased mitophagy, which led to the removal of damaged, ROS-producing mitochondria. Since mitochondrial ROS can induce inflammation, for example, by activating inflammasomes, such as NLRP3 [44], increased mitophagy might also be behind the observation of alleviated inflammation. However, further studies will be required to fully elucidate the role of NFE2L2/Nrf2, the precise pathways underlying autophagy/mitophagy activation, and the reduction in inflammation, all of which were outside the scope of this investigation. Furthermore, it should be kept in mind that these results were observed in an experimental cell model, using antimycin A and a comparatively large concentration of metformin, and more holistic studies in animal models or data from patients should be conducted to expand our understanding of the actions of metformin in AMD.

## 5. Conclusions

Metformin displayed anti-inflammatory and antioxidative properties in ARPE-19 cells. Studies with autophagy markers suggest enhanced activation of mitophagy upon metformin exposure, which could explain the detected positive effects on inflammation and mitochondrial oxidative stress.

## Figures and Tables

**Figure 1 cells-13-01433-f001:**
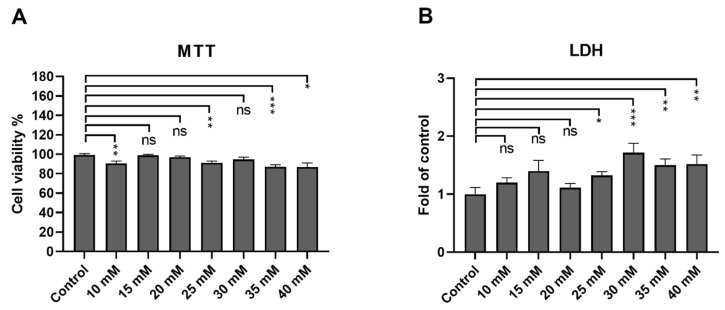
Metabolic activity (**A**) and LDH leakage (**B**) were measured following a 24 h metformin exposure. Cells were exposed to different metformin concentrations (10, 15, 20, 25, 30, 35, or 40 mM). Absorbance values of the exposed cells were compared to those of untreated control cells, the level of which was set to be 100% (**A**) or 1 (**B**). Data were combined from three experiments containing four parallel samples in each group per experiment (n = 12). Results are presented as mean ± standard error of the mean (SEM). *** *p* ≤ 0.0001, ** *p* ≤ 0.001, * *p* ≤ 0.05; ns = not statistically significant (Mann–Whitney U test).

**Figure 2 cells-13-01433-f002:**
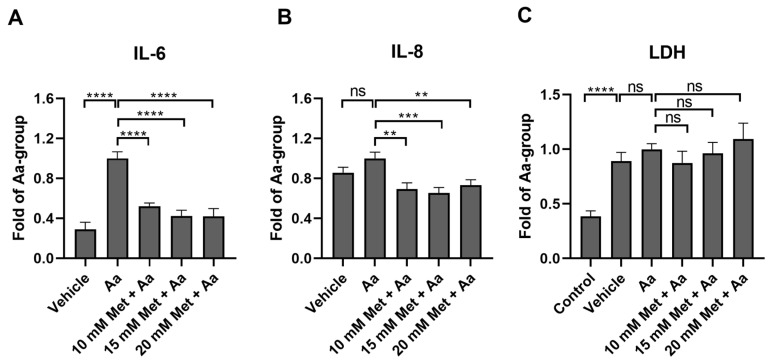
The effect of metformin on the levels of IL-6 (**A**), IL-8 (**B**), and LDH (**C**) after RPE cells were exposed to metformin (1 h, Met) and antimycin A (24 h, Aa). Aa was dissolved and diluted in DMSO, and an identical amount of DMSO served as a vehicle control. Data were combined from three experiments containing four parallel samples in each group per experiment (n = 12). Results are presented as mean ± SEM. **** *p* ≤ 0.00001, *** *p* ≤ 0.0001, ** *p* ≤ 0.001; ns = not statistically significant (Mann–Whitney U test).

**Figure 3 cells-13-01433-f003:**
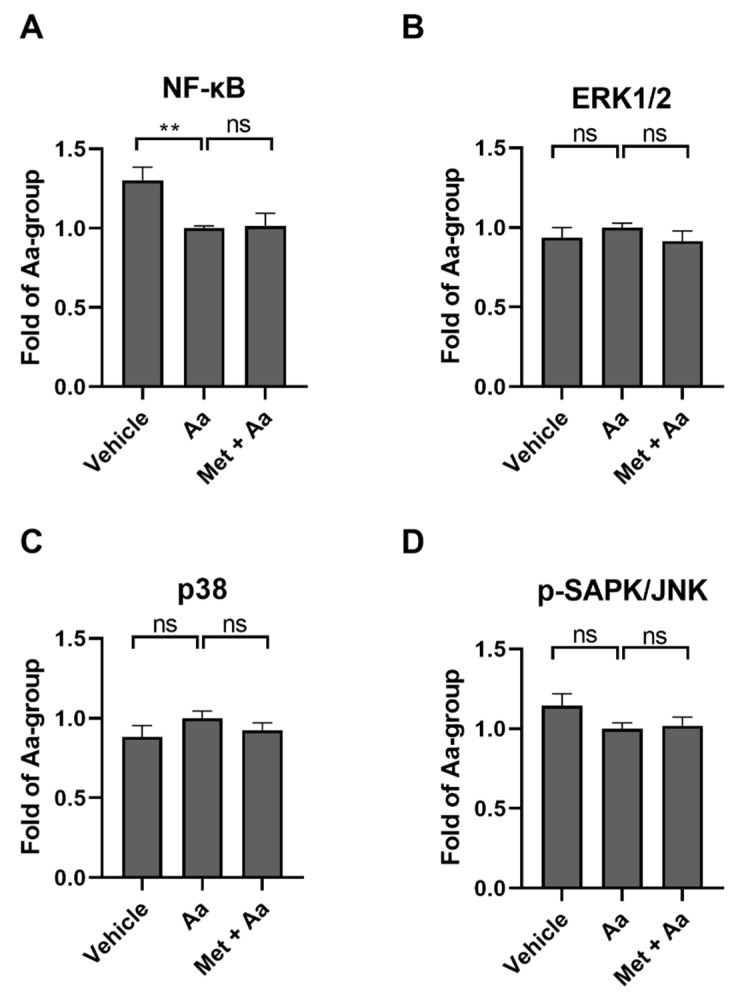
Potential mechanism behind the anti-inflammatory effects of metformin in cultured ARPE-19 cells. The levels of NF-κB (**A**) or phosphorylated ERK1/2 (**B**), p38 (**C**), and SAPK-JNK (**D**) after 1 h metformin (Met, 15 mM) and 2 h antimycin A (Aa, 10 µM) treatment. Aa was dissolved and diluted in DMSO, and an identical amount of DMSO served as a vehicle control. Data were combined from three experiments containing 2 (**A**) or 3 (**B**–**D**) parallel samples in each group per experiment (**A**, n = 6; **B**–**D**, n = 9). Results are presented as mean ± SEM. ** *p* ≤ 0.001; ns = not statistically significant (Mann–Whitney U test).

**Figure 4 cells-13-01433-f004:**
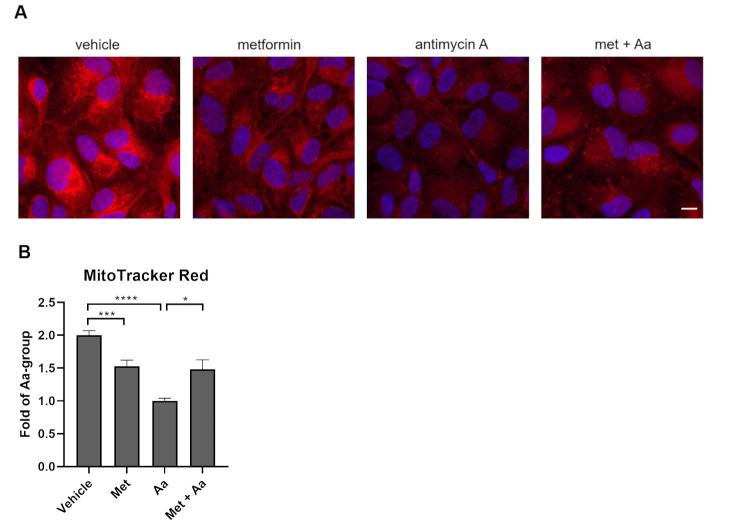
Mitochondrial membrane potential and mitochondrial quality were studied using the MitoTracker Red staining under a confocal microscope at 65× magnification after 1 h metformin (Met, 15 mM) and 24 h antimycin A (Aa, 10 µM) treatment (**A**). The relative brightness of mitochondria was calculated using the ImageJ software (**B**). Aa was dissolved and diluted in DMSO, and an identical amount of DMSO served as a vehicle control. Data were combined from three experiments containing 1 or 3 parallel samples in each group per experiment. Three cells were calculated per frame (n = 21). The scale bar equals to 10 µm. Results are presented as mean ± SEM. **** *p* ≤ 0.00001, *** *p* ≤ 0.0001, * *p* ≤ 0.05 (Mann–Whitney U test).

**Figure 5 cells-13-01433-f005:**
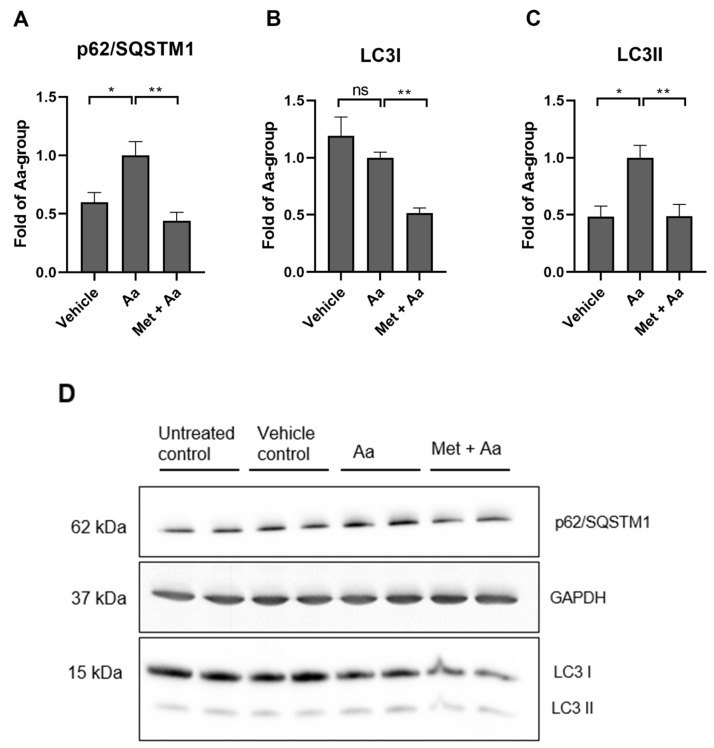
The levels of autophagy markers afterARPE-19 cells were exposed to metformin for 1 h (Met, 15 mM) and to antimycin A for 6 h (Aa, 10 µM). p62/SQSTM1 (**A**), LC3-I (**B**), and LC3-II (**C**) were measured using the Western blot technique, with a representative image being shown (**D**). Aa was dissolved and diluted in DMSO, and an identical amount of DMSO served as a vehicle control. Data were combined from three experiments containing 2 parallel samples in each group per experiment (n = 6). Results are presented as mean ± SEM. ** *p* ≤ 0.001, * *p* ≤ 0.05; ns = not statistically significant (Mann–Whitney U test).

**Figure 6 cells-13-01433-f006:**
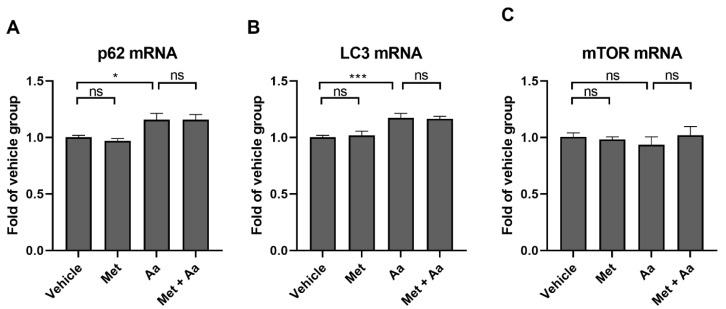
The messenger RNA levels of SQSTM1 (**A**), MAP1LC3B (**B**), and MTOR (**C**) were measured after 1 h metformin (Met, 15 mM) and 3 h antimycin A (Aa, 10 µM) treatment. Aa was dissolved and diluted in DMSO, with an identical amount of DMSO serving as a vehicle control. Data were combined from three experiments containing 3 parallel samples in each group per experiment (n = 9). Results are presented as mean ± SEM. *** *p* ≤ 0.0001, * *p* ≤ 0.05; ns = not statistically significant (Mann–Whitney U test).

**Figure 7 cells-13-01433-f007:**
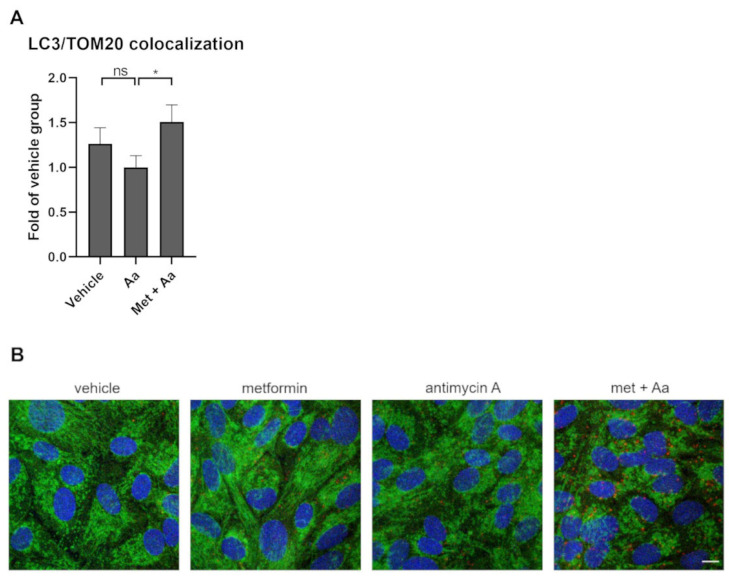
The colocalization of LC3 (red) and TOM20 (green) using Manders colocalization coefficient (**A**) was studied from images taken under a confocal microscope at 65× magnification (**B**) after ARPE-19 cells were exposed to metformin (Met, 15 mM) for 1 h and to antimycin A (Aa, 10 µM) for 24 h. Aa was dissolved and diluted in DMSO, with an identical amount of DMSO serving as a vehicle control. Data were combined from 4 experiments containing 3 parallel samples in each group per experiment. Three cells were calculated per frame (n = 21). The scale bar equals to 10 µm. Results are presented as mean ± SEM. * *p* ≤ 0.05; ns = not statistically significant (Mann–Whitney U test).

**Figure 8 cells-13-01433-f008:**
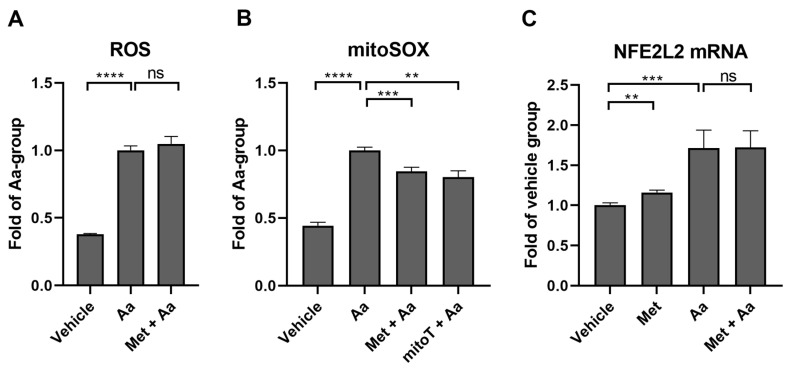
The levels of total cellular ROS (**A**) and mitochondrial ROS (**B**) after ARPE-19 cells were exposed to metformin (Met, 15 mM) for 1 h and to antimycin A (Aa, 10 µM) for 1 h (**A**), 2 h (**B**), or 3 h (**C**). Aa was dissolved and diluted in DMSO, with an identical amount of DMSO serving as a vehicle control. Data were combined from three experiments containing 6 parallel samples in each group per experiment (n = 18, (**A**)) or 4 parallel samples in each group per experiment (n = 12, (**B**)). Results are presented as mean ± SEM. **** *p* ≤ 0.00001, *** *p* ≤ 0.0001, ** *p* ≤ 0.001; ns = not statistically significant (Mann–Whitney U test).

**Table 1 cells-13-01433-t001:** Primer sequences used in qPCR analyses.

Gene	Forward	Reverse
*SQSTM1*	5′-GGA GCA GAT GAG GAA GAT CG-3′	3′-CTT CGG ATT CTG GCA TCT GT-5′
*MAP1LC3B*	5′-GCA GCA TCC AAC CAA AAT CC-3′	3′-CAT TGA GCT GTA AGC GCC TTC T-5′
*MTOR*	5′-AGC ATC GGA TGC TTA GGA GTG G-3′	3′-CAG CCA GTC ATC TTT GGA GAC C-5′
*NFE2L2*	5′-AAA TTG AGA TTG ATG GAA CAG CGA A-3′	3′-TAT GGC CTG GCT TAC ACA TTC A-5′
*GAPDH*	5′-GAT CAT CAG CAA TGC CTC CT-3′	3′-GGC CAT CCA CAG TCT TCT G-5′

## Data Availability

The data presented in this study are available upon request from the corresponding authors.

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
