# Peer review of "Metformin Alleviates Inflammation and Induces Mitophagy in Human Retinal Pigment Epithelium Cells Suffering from Mitochondrial Damage"

_cells, 2024, doi:10.3390/cells13171433_

Round 1
Reviewer 1 Report
Comments and Suggestions for Authors
In my opinion, the results and conclusions of this study may be biased by two factors not well considered by the Authors.
First, the concentrations of metformin adopted are much higher than usual plasmatic concentrations achievable in patients.
Second, Metformin is a well-known mitochondrial complex I inhibitor (as indirectly evident also in data showed -see fig 4-).
Any conclusive consideration cannot disregard these aspects.
Reviewer 2 Report
Comments and Suggestions for Authors
The abstract must include the applied methods and numerical data (concentrations, results expression). Also, the keywords should be different from those used in the title or abstract. The authors could consider exploring the Mesh terms available in the PubMed library.
The introduction section is adequate, but it would be beneficial if the authors included the main novelties that their hypothesis would present.
Proper references must be provided to support the metformin concentration; considering that the study is centered on the effects of this drug in the model, a concentration-curve effect would be adequate. The results present a concentration curve, but it was not properly described in the method section. Revise it.
Table 1 is a chart. Please revise it.
Incorrect citation in line 201;
Fig 5 depicts the WB images; The blots of the interaction group (Met+aa) are not optimal. The authors should present more representative images.
The authors are strongly encouraged to include a brief paragraph approaching the main limitations of their study for proper data interpretation. The possible perspectives should also be included.
Comments on the Quality of English LanguageRevise the text to polish the typo errors; numerous misspelled words exist. Please double-check the document.
Round 2
Reviewer 1 Report
Comments and Suggestions for Authors
The limitations of this study are now well stressed by Authors. So, I do not see further problems for publication
RS
Reviewer 2 Report
Comments and Suggestions for Authors
After this round of revision, my concerns were solved. The authors adequately answered the questions. Therefore, I am favorable to the acceptance of this manuscript.